# Innate attraction and aversion to odors in locusts

**Subhasis Ray**[1,2], **Kui Sun**[1], **Mark Stopfer**[1] *

**1** Eunice Kennedy Shriver National Institute of Child Health and Human Development, National Institutes of Health, Bethesda, Maryland, United States of America, **2** Plaksha University, Sahibzada Ajit Singh Nagar, Punjab, India

* stopferm@mail.nih.gov

## Abstract

Many animals display innate preferences for some odors, but the physiological mechanisms underlying these preferences are poorly understood. Here, with behavioral tests, we establish a model system well suited to investigating olfactory mechanisms, the locust *Schistocerca americana*. We conducted open field tests in an arena designed to provide only olfactory cues to guide navigation choices. We found that newly hatched locusts navigated toward, and spent more time near, the odor of wheat grass than humidified air. In similar tests, we found that hatchlings avoided moderate concentrations of major individual components of the food blend odor, 1-hexanol (1% v/v) and hexanal (0.9% v/v) diluted in mineral oil relative to control presentations of unscented mineral oil. Hatchlings were neither attracted nor repelled by a lower concentration (0.1% v/v) of 1-hexanol but were moderately attracted to a low concentration (0.225% v/v) of hexanal. We quantified the behavior of the animals by tracking their positions with the Argos software toolkit. Our results establish that hatchlings have a strong, innate preference for food odor blend, but the valence of the blend's individual components may be different and may change depending on the concentration. Our results provide a useful entry point for an analysis of physiological mechanisms underlying innate sensory preferences.

## Introduction

Animals are born with some innate sensory and behavioral preferences which develop further with age and experience. In humans, newborn females are attracted to odors they are exposed to immediately after birth [1], and neonates of both sexes can learn to associate an artificial odor with their mother in as little as one week [2]. Rabbit pups, in contrast to humans, receive little maternal care, so their survival after birth depends on their ability to successfully locate their mother's nipple to obtain milk. It has been shown that newborn rabbits use pheromonal cues for nipple-searching behavior without requiring any postnatal learning [3,4]. Insects such as locusts, whose eggs are laid in sand, do not usually hatch in contact with food; nor do they receive any parental care. So, one might hypothesize these animals must come equipped with innate sensory capacities enabling them to navigate to food, perhaps, at least in part, by tracking food odors. Most food odors consist of combinations of volatile chemicals emanating from

**Data Availability Statement:** The data underlying this article along with the code to reproduce the plots and statistics are available on Zenodo (doi: 10.5281/zenodo.7967047). The Argos toolkit (doi: 10.5281/zenodo.4653368) is free and open source,

and available on github: https://github.com/
subhacom/argos and https://github.com/
subhacom/argos_tracker.

**Funding:** S.R., K.S., and M.S. were funded by an
intramural grant from the Eunice Kennedy Shriver
National Institute of Child Health and Human
Development, National Institutes of Health, USA
(https://www.nichd.nih.gov/). The funders had no
role in study design, data collection and analysis,
decision to publish, or preparation of the
manuscript.

**Competing interests:** The authors have declared
that no competing interests exist.

the food. For example, wheat grass, which is eagerly eaten by locusts [5], releases at least 18 different volatiles. The major monomolecular volatile released by wheat grass is 1-hexanol (19.87 ± 2.5%) [6]. Humans typically perceive this chemical as smelling like freshly cut grass. Electrophysiological experiments have shown a wide range of concentrations of this odorant evokes strong neural activity in many projection neurons in the locust antennal lobes [7]. Hexanal is another major wheat grass volatile (2.55 ± 0.14%) [6] which has also been identified as a component of the aggregation pheromone blend in a closely related locust species, *Schistocerca gregaria* [8].

A substantial literature documents chemicals attractive or repellent to locusts. Locusts are generalist foragers and consume a wide variety of plants [9–11]. In *Schistocerca gregaria*, fifth instar nymphs raised on other plants are attracted to wheat grass [12]. Many of the volatiles present in insect food are also found in insect feces, and some of these chemicals or their derivatives may function as pheromones. For example, fifth instar nymphs of *S. gregaria* have been reported to show strong aggregation responses to a synthetic blend mimicking fecal volatiles including the aldehydes hexanal, octanal, nonanal, and decanal, as well as their corresponding acids [8]. Two-day-old males of the oriental migratory locust (*Locusta migratoria manilensis)* also exhibit aggregation responses when presented an odor blend containing hexanal and other fecal volatiles [13]. Among various chemicals produced by gregarious adults of *Locusta migratoria* 4-vinylanisole attracts, but phenylacetonitrile repels conspecifics [14].

Here we investigated the intrinsic preferences of newly hatched locusts for a complex food odor. Locust hatchlings just emerging from eggs buried in sand were led into an open field arena that allowed for the controlled presentations of odorants. We monitored the movements of the hatchlings within the arena by using the software toolkit Argos [15] which records videos and tracks the locations of multiple animals throughout experiments.

In insects, the antennae serve as the primary olfactory organs. However, insects can also possess olfactory receptors in their mouth parts and, in some species, their ovipositors [16–19]. To evaluate the contribution of the locust antennae to odor-directed navigation, in some experiments we anesthetized the hatchlings by cooling them and then amputated their antennae. In adult locusts, cooling the antennae leads to a reversible reduction in both spontaneous and odor-elicited activity in olfactory receptor neurons [20].

## Materials and methods

### Behavior experiments

American desert locusts (*Schistocerca americana*) were reared in our laboratory at 37˚C in a 12 h light and 12 h dark cycle in crowded colonies (300–400 animals in 45cm x 45 cm x 45 cm screened cages). Our locusts were fed fresh oat bran and wheatgrass (*Triticum aestivum*) grown from seeds in the laboratory under a 12 h light and 12 h dark cycle at 37˚C. The adult females laid eggs in sand-filled plastic cups which were cleaned of plant and other matter and were kept in an incubator at about 29˚C until hatchlings began to appear. The phase of newly hatched locusts is indeterminate. Initially green, a characteristic of the solitary phase, our locusts typically turn brown, a characteristic of the gregarious phase, by around the 3rd instar. Phase transitions can occur at multiple time scales; individuals can switch within hours, but epigenetic changes contributing to polyphenism can develop and accumulate over several generations [21]. Our locust hatchlings are progeny of many generations of gregarious locusts.

Cups with fresh hatchlings were fit with a plastic test tube through the lid and wrapped with a black cloth (Fig 1). Locusts tend to climb upwards (negative geotaxis) and towards light (positive phototaxis), and our hatchlings spontaneously climbed up into the test tube. When 10–20 hatchlings had climbed up, the test tube was removed and affixed to the entrance of the arena

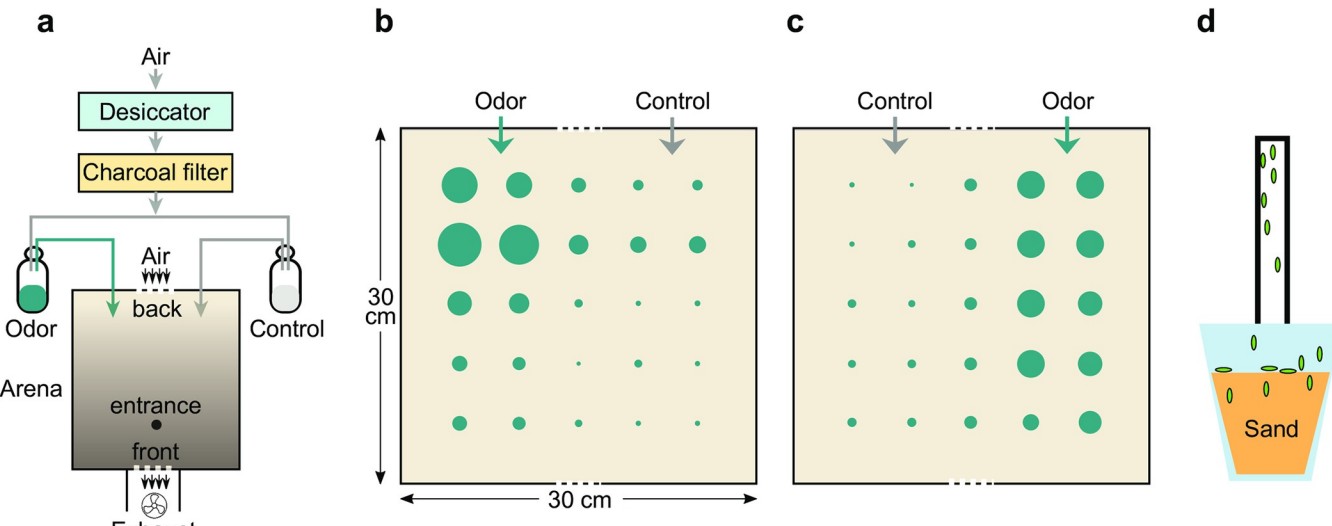

**Fig 1. Experimental apparatus for testing odor preferences of freshly hatched locusts.** (a) Schematic of the setup for testing odor preferences of hatchlings; (b) and (c) normalized odorant concentration measured by a mini Photo-Ionization Detector at different locations of the arena when delivered through the left and the right port, respectively. Circle diameter indicates relative concentration. (d) Technique for collecting hatchlings without touching them.

from below. The entrance was initially covered with a small petri dish. To start the experiment, the petri dish was removed, allowing hatchlings to enter the arena. This procedure eliminated the need to handle the hatchlings before testing their behavior. All experiments were conducted inside our laboratory at room temperature with artificial light.

In all experiments we made two measures to compare the attraction of hatchlings to the odorized and non-odorized port: (1) the number of times a hatchling entered semicircular regions of interest (ROI) around the ports; and (2) the total amount of time it spent within the ROIs.

In our experiments multiple animals (~10–20) were delivered to the arena entrance, and, in a typical experiment, about half would walk to at least one of the ROIs within the test period (usually 100 min). Only hatchlings entering an ROI were included in our analysis. We tested preference for grass-juice compared to humid air in 24 experiments with 316 hatchlings, 1% hexanol vs. mineral oil in 19 experiments with 223 hatchlings, of which 123 walked to at least one port; 0.1% hexanol vs. mineral oil in 14 experiments with 146 hatchlings, of which 83 walked to at least one port; 0.9% hexanal vs. mineral oil in 12 experiments with 138 hatchlings, of which 115 walked to at least one port; and 0.225% hexanal vs. mineral oil in 12 experiments using 134 hatchlings, of which 95 walked to at least one port.

To test whether the antennal olfactory system was needed to indicate olfactory preferences within the arena, in some experiments we removed the antennae. Hatchlings were collected in a test tube and anaesthetized by cooling in ice. Then, under a dissection microscope, both antennae were removed with a pair of sharp scissors and antenna stumps were coated in wax. Hatchlings then recovered at room temperature for about 30 minutes in a test tube before being introduced to the behavior arena as described above. In these 12 experiments we used 131 hatchlings, and the tracks of the 35 hatchlings which walked to at least one of the odor sources were analyzed.

To test whether cooling the hatchlings affected their odor preference, some were cooled in ice as above, but their antennae were left intact. These hatchlings were also allowed to recover at room temperature for about 30 minutes before they were introduced to the arena. A total of 59 tracks of hatchlings visiting either ROI were analyzed from 22 experiments with 201 hatchlings.

After each trial in the two-choice behavior tests, the inside of the arena was wiped with 70% ethanol and allowed to dry completely. A new batch of hatchlings was used for each trial.

## Behavior arena

The behavior arena (Fig 1) consisted of a clear acrylic tray (30.5 cm x 30.5 cm x 5 cm) with a non-reflective glass (33 cm x 33 cm, True Vue) cover (Fig 1A). To reduce visual clutter from reflections, the inner walls of the tray were covered with window film. Air was drawn through the arena through two nylon mesh-covered slots on opposite walls by a computer fan (Comair Rotron 25 mm x 10 mm 12V DC) powered by a DC adapter with an adjustable voltage output. We used a smoke test to visualize and calibrate the airflow, adjusting the fan's supply voltage to minimize turbulence, resulting in an air speed of about 0.35 m/s in front of the exhaust slot.

The arena was illuminated from below by a fluorescent tube light placed near the back of the arena. The bottom of the arena was covered with white paper on the outside, which diffused the light and created a gradient so the back of the arena was brighter than the front. After entering near the front of the arena, hatchlings tended to phototax towards the back. A cellphone based light meter confirmed the left and right sides near the back of the arena were equally bright.

Two 50 ml glass bottles were fitted with airtight silicone plugs pierced by two polyethylene tubes (Fig 1A). Desiccated, activated carbon filtered air flowed into each bottle through one tube and out through the other. Output tubes were inserted into the behavior arena through holes drilled on the airflow-entry wall, opposite from the exhaust fan. Odor and control ports were alternated between experiments.

To determine the distribution of odorant within the arena under experiment conditions, we used a photo-ionization detector (PID, Aurora Scientific) to measure odorant intensity in a 5 cm square grid (Fig 1B). 100% 1-hexanol was delivered through the port on one side as described above, with the glass cover in place. A polyethylene tube attached to the PID was inserted through the entrance (Fig 1A) and placed at a grid point, and the PID signal was recorded thrice for 10 s at 20 s intervals. The baseline signal of the PID was also measured without the odorant. Fig 1B and 1C show, for odor delivered on the left and the right respectively, the time-averaged odor signal from the second trial, normalized by the time-averaged baseline signal [(odor–baseline) / (max(odor)–baseline)]. These measurements confirmed that odorants were laterally distributed, as desired.

## Odorants

To test responses to a food odor, we used wheat grass (*Triticum aestivum*) grown in our laboratory. Fresh grass was crushed with a mortar and pestle, and the juice was then pressed through a cell filter mesh. 1 ml extracted juice was placed as a test odorant in one bottle, and 1 ml deionized water in the other bottle served as control. Bottles were thoroughly cleaned and dried between trials. Grass juice was used for testing innate attraction of undisturbed hatchlings as well as for evaluating the olfactory capabilities of locusts without antennae, and those with intact antennae but cooled and then revived to room temperature.

To test monomolecular odorants, we used 1-hexanol (Sigma), the dominant component of grass juice volatiles, and hexanal (Sigma), which is also a significant component of grass juice [6]. These odorants were diluted in mineral oil, and pure mineral oil was used as the control. All concentrations are reported as volume / volume. We tested 1-hexanol at 1% and 0.1% concentrations, and hexanal at 0.9% and 0.225% concentrations.

## Behavior tracking

Locust behavior in the arena was recorded by a USB web camera (Logitech Pro Stream C922x) in 100-minute segments using the Capture utility of the Argos software toolkit [15]. We then used this toolkit to analyze video segments offline, keeping only time-stamped frames that included detectable motion. We then processed these reduced videos with the Argos Tracking tool to automatically track hatchling movements. To accurately map tracks in all videos to the same coordinates, we manually marked four corners of the arena floor in each video, and then based on these points, computed the transformation matrix from video coordinates to world coordinates, and then, using this matrix, transformed video coordinates into world coordinates. Finally, because the odor and control ports were alternated between trials, we brought them into the same alignment by flipping the coordinates left-right as necessary. Afterwards, the timings of hatchling movements were reconstructed from the timestamp associated with each video frame.

## Data analysis

All data analysis was carried out by custom scripts written in Python using Python-scipy stack including the Pandas library.

We used t-tests to compare stay times (total time a hatchling spent in an ROI) and ROI entries (total number of times a hatchling crossed into an ROI) between odor and control sides. Although none of the differences were normally distributed (confirmed by Shapiro-Wilk tests), the use of t-tests was appropriate because n > 30 for all comparisons, and thus the central limit theorem could be applied. We also conducted nonparametric Wilcoxon signed rank tests which produced p values (not shown) supporting the same conclusions reported here.

To compare attraction toward the test odor port and the control port over time we used the lifelines module in Python for survival analysis with Kaplan-Meier fit. The Kaplan-Meier curve is a step function indicating the probability of "survival," i.e. the probability that an event of interest has not yet occurred. In the experiments comparing attraction towards grass juice odor and control, we computed, for each hatchling, the interval between first detecting the hatchling in the arena and its first entrance to the ROI around the odor port. We used these intervals to make a survival plot, with the entry into the ROI as the event of interest (orange line in Fig 2D). Animals which did not enter the ROI within the experiment duration of 100 minutes were "censored." Similarly, the first entry into the ROI around the control port was the event of interest for control, and the time interval from first detecting the hatchling until this event was used in the Kaplan-Meier fit (blue line in Fig 2D). The log-rank test was used to compare these curves. Note that here we are comparing the first entry into the odor ROI with that into the control ROI, and an animal entering the two ROIs is not mutually exclusive; if a hatchling visited both ROIs, then it is included in both curves with the corresponding time intervals. For example, an animal could first enter the odor ROI at time *t1*, and then move into the control ROI at time *t2*. Then *t1* would be included in the survival plot for odor, and *t2* would be included in the survival plot for control.

## Results

### Naïve locusts are attracted to food odor

To determine whether locusts are innately attracted to food odor, we conducted a two-choice test between grass juice odor and water vapor. We used newly hatched locusts that had no prior exposure to food or food odors. In this first instar stage the hatchlings have not yet developed wings and move by walking or jumping. Fig 2A shows overlaid tracks from all hatchlings in all experiments, and S1 Fig shows sample tracks of randomly selected animals. Track color

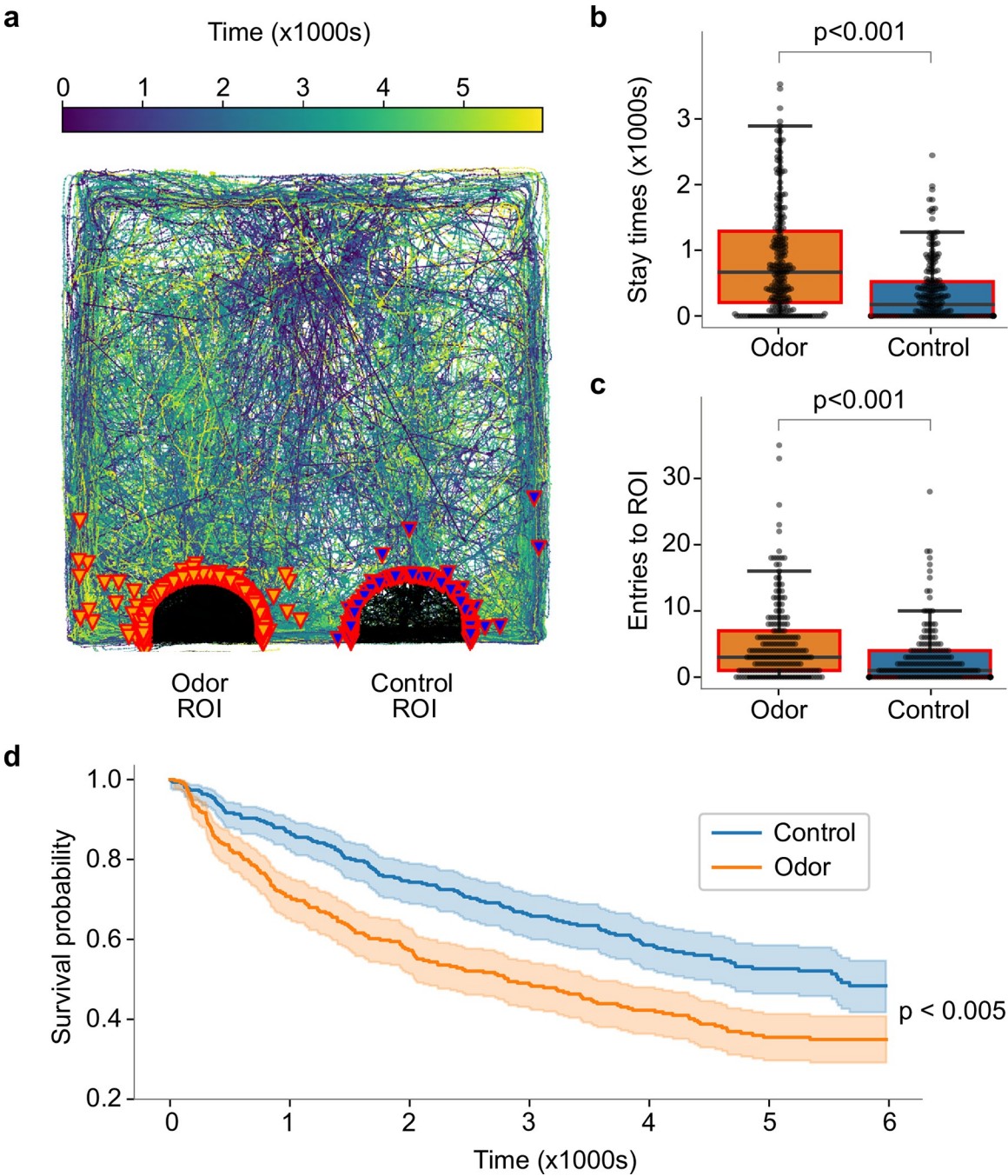

**Fig 2. Hatchlings move more quickly to, spend more time in, and make more visits to the area around the source of grass juice odor.** (a) Overlaid tracks of all hatchlings in a test of attraction to wheat grass juice odor. Purple/dark blue–early part of the track, light green/yellow–late part of the track. Portions of tracks within region of interest (ROI) marked in black, orange arrow heads indicate hatchlings crossing into the region of interest around grass juice and magenta arrow heads indicate hatchlings crossing into the region of interest around control. (b) Hatchlings stay within the ROI around grass juice (orange) longer than that around control (blue). (c) The number of entries into the odor ROI is also larger than control, indicating that the hatchlings cross into it more often. (d) Kaplan-Meier survival plot, where events are defined as first entries into the region of interest around odor (orange) or control (blue), shows hatchlings quickly head toward the odor port.

in these images was set to change from dark purple to blue to green to yellow over time, showing that many hatchlings moved directly towards the source of grass juice odor after entering the arena (S1 Fig). To quantify the attraction of the hatchlings to the food odor, we computed the amount of time they spent in semicircular regions of interest (ROI) of 4 cm radius around the odor and control ports. Hatchlings spent significantly more time near the odor port than the control port (Fig 2B; paired t-test, n = 213, p = 6.44e-11). We also found the hatchlings crossed into the food odor port ROI significantly more times than into the control port ROI (Fig 2C, paired t-test, n = 213, p = 1.16e-6), indicating that they returned to the food odor source repeatedly while exploring the arena. We computed a preference index (PI) based on total stay time in the ROIs as $(\Sigma t_{Odor} - \Sigma t_{Control}) / (\Sigma t_{Odor} + \Sigma t_{Control})$, for sums over the entire population, of PI = 0.41. We also computed a PI for the number of ROI entries using the same formula, replacing time $t$ with the number of ROI entries $n$, and this yielded PI = 0.32. Both PIs indicate the hatchlings move preferentially toward grass juice odor.

Finally, we applied a survival analysis using the Kaplan-Meier (KM) method to determine the rates at which hatchlings moved toward odor or control ports (log-rank test statistic = 20.17, p < 0.005). In this statistic, hatchlings are removed from the "survival" pool when they first enter either ROI, so survival curves shown in Fig 2D indicate the time it took for each hatchling to choose. The KM plots show hatchlings entered the food odor ROI significantly faster than the control ROI. Also, the slope of the curve for the odor is steeper at the beginning, indicating that the hatchlings tend to move quickly towards the odor port upon entering the arena. Because no visual or other cues distinguished the source of the food odor from that of the control, this result establishes that hatchlings could use odor guided navigation to reach the source of an attractive food blend odor.

## Attraction to food odor is mediated by antennae

Antennae are the main olfactory organs in insects, but locusts also have odorant receptors on their mouth parts. To test whether the antennal olfactory pathway was necessary for the hatchlings' innate attraction toward food odor, we conducted behavioral experiments identical to those described above but with hatchlings whose antennae had been removed while they were anesthetized by cooling. We found hatchlings lacking antennae showed no preference for the food odor port compared to the control port (Fig 3A–3C; two-tailed paired t-test, for stay time n = 35, p = 0.44, PI = -0.10, and for number of ROI entries n = 35, p = 0.57, PI = 0.07, showing no significant difference). We noted that these antennectomized hatchlings, unlike intact ones, tended to walk in looping arcs while heading toward the back of the arena (S2 Fig), possibly drawn there by the brighter light, or by air movements.

To test for possible effects of anesthesia by cooling, we also conducted control experiments in which hatchlings were cooled over ice, but their antennae were left intact. These animals, after recovering at room temperature, were, by most measures, significantly attracted towards food odor (Fig 3D–3F; paired t-test, for stay-time n = 59, p = 0.08, PI = 0.26, and for the number of ROI entries n = 59, p = 0.002, PI = 0.38).

Together, these results indicate that the antennal olfactory pathway primarily mediates the hatchlings' naïve attraction toward food odor.

## Naïve hatchlings avoid a single component of food odor blend

We next asked whether single components of grass odor attract the hatchlings. We therefore conducted two-choice tests as above but replaced the food blend odorant with 1-hexanol diluted in mineral oil (1% v/v), tested against pure mineral oil. Notably, the hatchlings were not attracted to the dilute 1-hexanol, and instead showed a significant tendency to avoid it

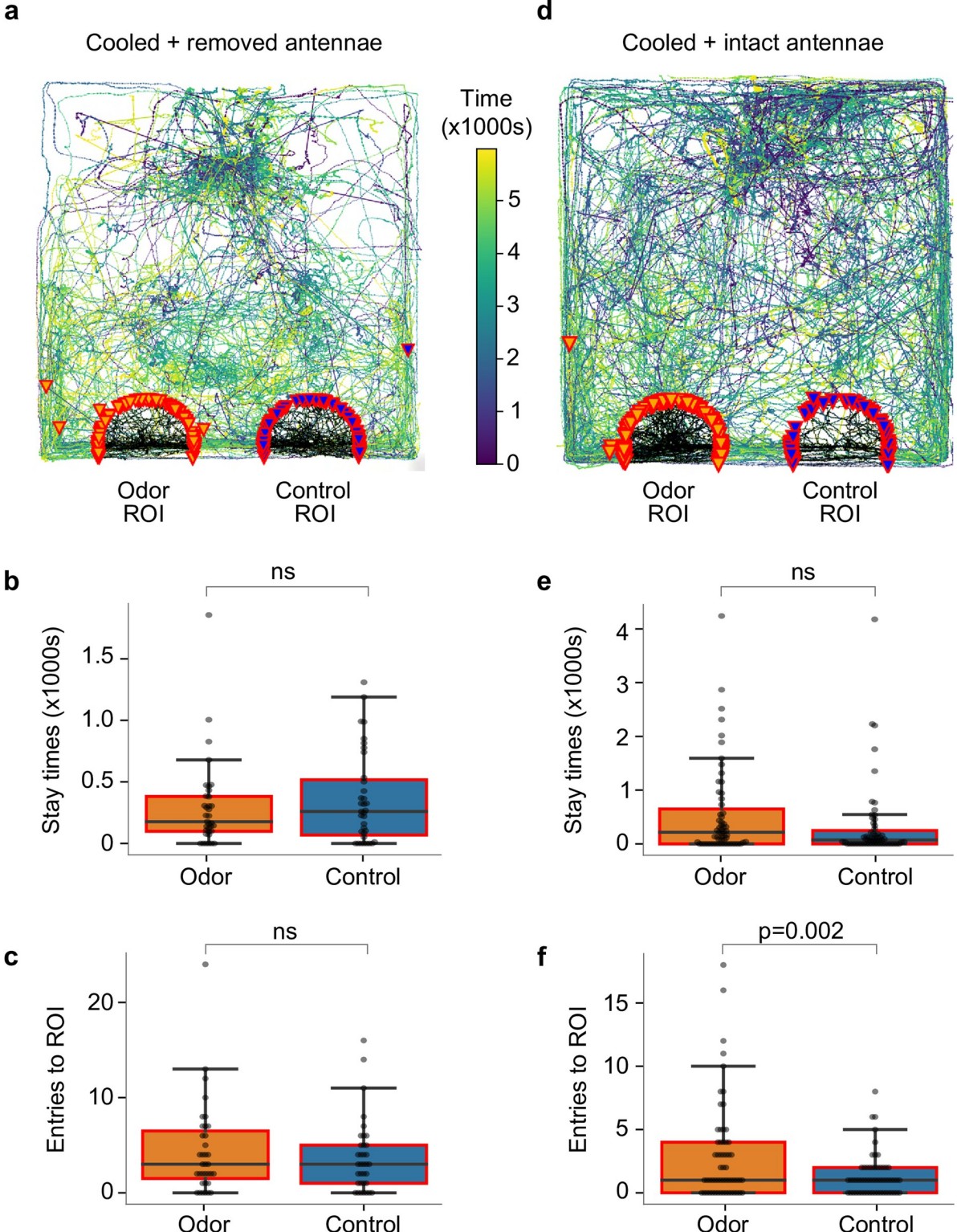

**Fig 3. Hatchlings with their antennae removed do not show any difference in attraction towards grass juice odor and control.** (a) Overlaid tracks from all antennectomized hatchlings; notably, some walked in circular patterns. (b) Stay time in control region of interest and odor region of interest, (c) number of entries into control region of interest and odor region of interest. Control experiments revealed that anesthesia by cooling does not by itself affect odor preference nor navigation to a preferred odor source when antennae are intact: (d) overlaid tracks of all the hatchlings from these control experiments, (e) stay time in control region of interest and odor region of interest; results fell just short of statistical significance, (f) number of entries into control region of interest and odor region of interest.

(Fig 4A): hatchlings spent significantly less time within the ROI around the odor port than the control port (Fig 4B, paired t-test: n = 123, p = 3.96e-4, PI = -0.38), and made fewer entries to the odor port than the control port (Fig 4C, paired t-test: n = 123, p = 9.05e-5, PI = -0.54).

To test whether this avoidance behavior was caused by the high intensity of the test odor, we next reduced the concentration of 1-hexanol in the odor bottle by an order of magnitude (0.1% v/v). However, this concentration of 1-hexanol elicited neither attraction nor repulsion from hatchlings, which showed no difference in affinity for the odorant or the control (Fig 4D–4F). In this case the PI for stay time was -0.02 and that for the number of ROI entries was -0.06. Two-tailed paired t-tests showed no differences (n = 87, p = 0.86 for stay time; n = 87, p = 0.60 for number of ROI entries). Together, these results indicate hatchlings were not attracted by 1-hexanol alone.

## Attraction to an aggregate pheromone depends on its concentration

Another major component of wheat grass juice is hexanal. We found odor from dilute hexanal (0.9% v/v in mineral oil) repelled hatchlings (Fig 5A–5C; n = 115, paired t-test, p = 2.93e-4, PI = -0.27 for stay-time and p = 1.80e-4, PI = -0.34 for number of ROI entries). Notably, further diluted hexanal (0.225% v/v) attracted hatchlings as shown by significantly longer stay times near its outlet (Fig 5D–5F; n = 95, paired t-test, p = 0.038 and PI = 0.14), though the difference in the number of entries into ROI fell short of statistical significance despite showing a positive preference index (p = 0.15, PI = 0.21). Together, these results demonstrate that even an etholo-gically important attractive odor can change valence when presented at different concentrations.

## Control for mineral oil

Finally, to test whether possible odors from the mineral oil used as control in these experiments might have affected the movements of hatchlings, we repeated the 2-choice test with air passing through a clean, empty bottle tested against air passing through a bottle containing mineral oil. Hatchlings showed no preference for either choice in these experiments in terms of stay time (n = 81, with two-sided paired t-test p = 0.31; PI = 0.10), or the number of ROI entries (n = 81, with two-sided paired t-test p = 0.29; PI = 0.09) (Fig 6A–6C), indicating the hatchlings were neither attracted nor repelled by mineral oil.

## Discussion

Here we found naïve, newly hatched locusts are innately attracted by the odor of wheat grass, a plant readily consumed by locusts of all ages ([5]; our own unpublished observations). We also found that, to navigate towards the odor source, hatchlings require their antennae. What mechanism could explain this innate attraction? One possibility is hatchlings innately overex-press olfactory receptors sensitive to plant volatiles, allowing them to find plants by moving toward anything emitting an odor that elicits a strong antennal signal. This seems unlikely because animals are generally repelled by strong odors ([22,23]; also, see Figs 4 and 5); we are testing this idea now. Another possibility is that genetically regulated hard-wiring links anten-nal lobe projection neurons (PNs) or glomeruli responding to food odors with neuronal path-ways that assign positive valence. This, too, seems unlikely because many locust PNs with projections distributed throughout the antennal lobe respond to both food odors and non-food odors [7]. Could learning play a role? [3] demonstrated one-shot associative learning in rabbits: pups exposed just once to an artificial odor while suckling later searched for nipples on a female cat scented with the artificial odor. Could the locust hatchlings experience one-shot learning for food odors left alongside eggs laid by the mother? This scenario seems

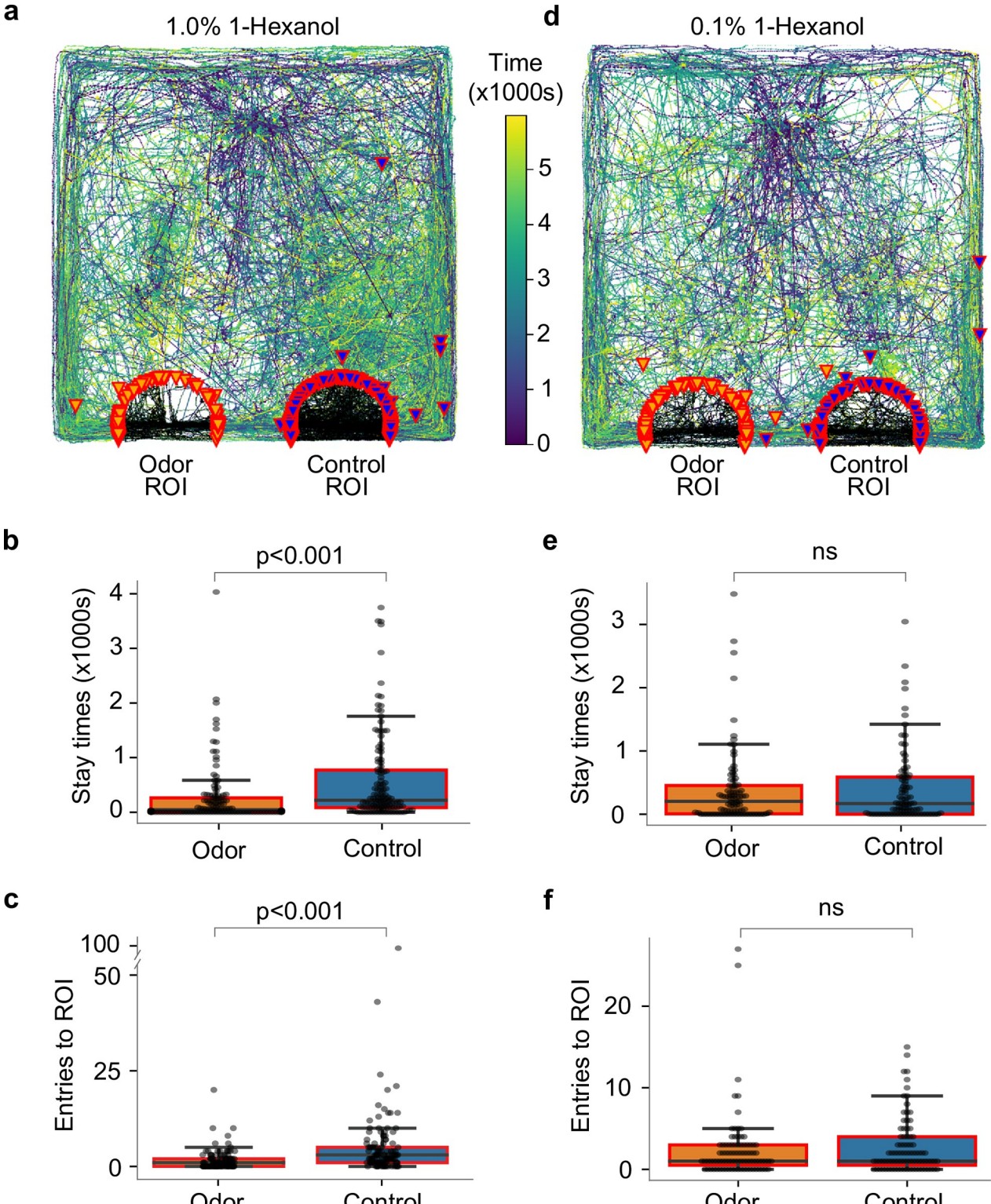

**Fig 4. Naive hatchlings avoid 1-hexanol.** (a) Overlaid tracks of hatchlings tested for attraction towards 1% 1-hexanol compared to mineral oil. (b) Stay-time of hatchlings in the region of interest around 1-hexanol compared to that around control. (c) Number of region of interest entries. (d) Overlaid tracks of hatchlings tested with 0.1% 1-hexanol. (e) The stay-time in the odor region of interest was not significantly different from that of control. (f) Number of region of interest entries did not show significant differences between odor and control, either.

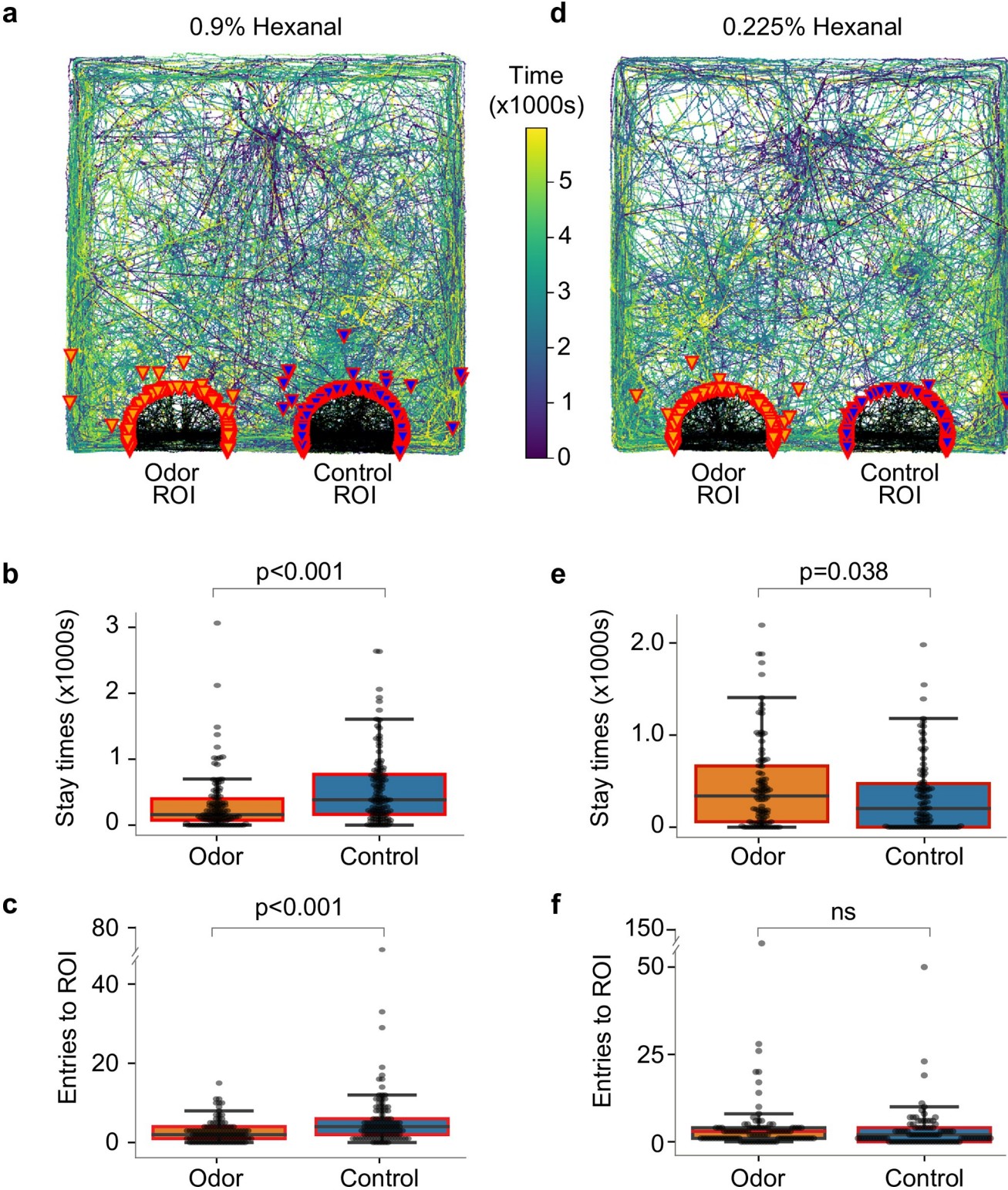

**Fig 5. Naive hatchlings avoid a high concentration of hexanal but are attracted to a lower concentration.** (a) Overlaid tracks for 0.9% hexanal, (b) comparison of stay times in the region of interests, and (c) number of entries into the region of interests. (d) Tracks for 0.225% hexanal, and comparison of (e) stay times and (f) number of region of interest-entries.

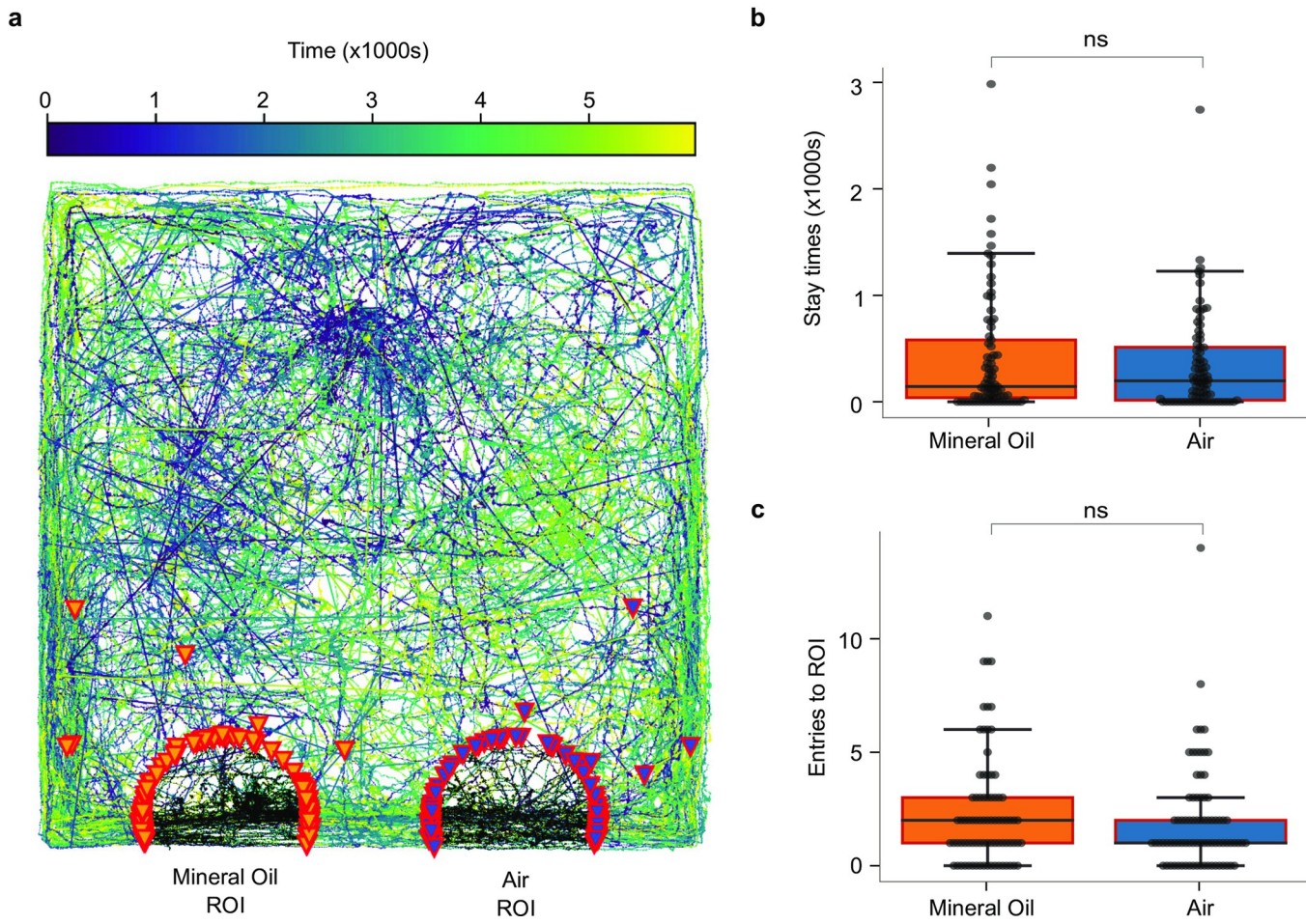

**Fig 6. Control experiment comparing attraction to mineral oil to desiccated, filtered air.** (a) Overlaid tracks of all the hatchlings, (b) comparison of stay time in the region of interests, and (c) comparison of the number of region of interest entries. None of these measures showed any significant differences between filtered, desiccated air and mineral oil.

unlikely because it includes no association between possible food odors and a feeding event. But a more likely possibility is that olfactory wiring downstream from the antennal lobe is shaped during development by volatile components of the food consumed by the mother, and then deposited in egg pods. Consistent with this possibility are cases in which chemical components of food, or metabolic products derived from them, function as pheromones [24], potentially priming development. Also, odor preferences of newborn mice can be influenced by the diets of their mothers [25]. In locusts, this possibility could be tested by manipulating the mother's diet or the volatiles present in egg pods. One could, for example, rear a generation of locusts on a synthetic diet [26] to avoid the deposition of plant material along with the egg pods. Afterwards, offering hatchlings from such egg pods a choice of grass juice volatiles or odorants present in the synthetic food could reveal whether the presence of chemicals in the mother's diet is responsible for the preference of the hatchlings for these chemicals.

Insects require their antennae to navigate toward odor sources, as demonstrated by experiments employing antenna amputation in honeybees [27] and cockroaches [28]. Experiments with cockroaches suggested that the palp olfactory system may be engaged primarily for odorants at very short distances, whereas longer distances require the antennal olfactory system [28]. Antennectomy has been shown to reduce overall locomotion in 5th instar locusts and to

impair their tendencies to walk upwind and toward odors, although palp-mediated olfaction may, in this developmental stage, enable some odor-guided movement [29]. Consistent with this earlier work, our experiments with antennectomized locust hatchlings also revealed reduced and atypical locomotion; however, we found no evidence for palp-mediated olfactory guidance. To anesthetize hatchlings, we briefly cooled them. Cooling has also been shown to reversibly reduce spontaneous and odor-elicited activity in olfactory receptor neurons and to increase behavioral sensitivity to odors in insects [20,30,31]. In this study we did not specifically investigate cooling-induced changes in olfactory sensitivity, but we observed that in animals with intact antennae that had been cooled, olfactory capabilities recovered fully upon warming to room temperature.

Our results show that even relatively low concentrations of 1-hexanol (1% v/v diluted in mineral oil), a major monomolecular component of wheat grass that is known to elicit widespread neuronal responses in the locust olfactory pathway, evokes avoidance behavior in naïve locust hatchlings, while lower concentrations (0.1%) elicited neither attraction nor avoidance. This result is consistent with earlier findings in other model systems showing the importance of background odors or odor blends in determining valence. The black bean aphid, for example, is repelled by many of the individual volatile compounds of its host plant, though attracted to their blend [32]. The Asian tiger mosquito is attracted to human body odor, but not to its individual components [33]. And, the moth *Manduca sexta* is not attracted by single or small subsets of components present in its favored food source, the *Datura wrightii* flower, but is attracted by larger synthetic mixtures of its major components [34]. Moreover, the odor of the *Datura* flower combined with that of the leaf of this plant elicits a stronger positive behavioral response than the floral odor alone [35].

Animals may have different behavioral thresholds for navigation and consummatory behaviors. For example, fifth instar locusts will display a palp opening response (POR) when many types of odorants in filter paper are presented, including hexanol and hexanal at concentrations avoided in our locomotion experiments [36]. Olfactory neurons in locusts have been shown to respond to wide ranges of concentrations of odorants (for example, to 0.1% to 100% 1-hexanol dissolved in mineral oil [7,37]). To further elucidate the neural basis of innate olfactory behaviors, it will be important to investigate a wider range of odor concentrations.

## Supporting information

**S1 Fig. Sample tracks of locusts with intact antennae in two-choice test between grass juice odor and humid air.** Each row shows individual animals from one experiment. Orange circle indicates the odor ROI and the blue circle the control ROI. Tracks are color-coded by time. (TIF)

**S2 Fig. Sample tracks of antennectomized locusts in the two-choice test between grass juice odor and humid air.** Each plot is an individual animal from a different experiment. Orange circle indicates the odor ROI and the blue circle the control ROI. Tracks are color-coded by time. (TIF)

## Acknowledgments

We thank NIMH Section on Instrumentation for help with the experimental setup, Diantao Sun for animal care, and members of the Stopfer lab, especially Zane Aldworth for helpful comments and suggestions. We also thank Erica Varga for assisting with reviewing some of the data.

## Author Contributions

**Conceptualization:** Subhasis Ray, Kui Sun, Mark Stopfer.

**Data curation:** Subhasis Ray, Kui Sun.

**Formal analysis:** Subhasis Ray.

**Funding acquisition:** Mark Stopfer.

**Investigation:** Subhasis Ray, Kui Sun.

**Methodology:** Subhasis Ray, Kui Sun.

**Project administration:** Mark Stopfer.

**Resources:** Mark Stopfer.

**Software:** Subhasis Ray.

**Supervision:** Mark Stopfer.

**Visualization:** Subhasis Ray.

**Writing – original draft:** Subhasis Ray.

**Writing – review & editing:** Subhasis Ray, Mark Stopfer.

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
