## [Decision Letter · Decision Letter 0]

25 Apr 2023

PONE-D-23-10041Innate attraction and aversion to odors in locustsPLOS ONE

Dear Dr. Stopfer,

Thank you for submitting your manuscript to PLOS ONE. After careful consideration, we feel that it has merit but does not fully meet PLOS ONE’s publication criteria as it currently stands. Therefore, we invite you to submit a revised version of the manuscript that addresses the points raised during the review process. Please submit your revised manuscript by Jun 09 2023 11:59PM. If you will need more time than this to complete your revisions, please reply to this message or contact the journal office at plosone@plos.org. Please include the following items when submitting your revised manuscript:A rebuttal letter that responds to each point raised by the academic editor and reviewer(s). You should upload this letter as a separate file labeled 'Response to Reviewers'.A marked-up copy of your manuscript that highlights changes made to the original version. You should upload this as a separate file labeled 'Revised Manuscript with Track Changes'.An unmarked version of your revised paper without tracked changes. You should upload this as a separate file labeled 'Manuscript'.If applicable, we recommend that you deposit your laboratory protocols in protocols.io to enhance the reproducibility of your results. Protocols.io assigns your protocol its own identifier (DOI) so that it can be cited independently in the future. For instructions see: https://journals.plos.org/plosone/s/submission-guidelines#loc-laboratory-protocols. Additionally, PLOS ONE offers an option for publishing peer-reviewed Lab Protocol articles, which describe protocols hosted on protocols.io. Read more information on sharing protocols at https://plos.org/protocols?utm_medium=editorial-email&utm_source=authorletters&utm_campaign=protocols.

We look forward to receiving your revised manuscript.

Kind regards,

Efthimios M. C. Skoulakis, PhD

Academic Editor

PLOS ONE

Journal Requirements:

Additional Editor Comments:Although both reviews are positive there are a number of comments raised by Reviewer 2 that need to be thoroughly addressed, aiming to improve readability of teh manuscripts and be clear on all protocols and procedures used. these comments are summarized below:

The current study has a good sample size, and I was very intrigued and impressed with the variety of observations that the authors have made. However, many details are missing and unclear in the abstract, introduction, and methods and it is not obvious what the authors have studied until the result section. Moreover, information is scattered in the method and therefore better organization in the method section is required. It was hard to understand how the tracking device works (which is the main observation tool used in the study) so a detailed explanation in the introduction/method sections is required. Finally, the authors need to mention whether the locusts are in solitarious or gregarious phases as the odor capacity changes between the two phases. Below are detailed reviews for each section.

**Abstract**

Need to mention whether it is gregarious or solitarious and whether lab-reared or wild-caught.
“*We conducted open field two-choice tests with purely olfactory stimuli. In these tests, newly hatched locusts navigated toward, and spent time near, the source of a food odor blend, crushed wheat grass*”

What do you mean by purely olfactory stimuli? Do you mean by synthetic compounds? Spent time near wheat grass more than…? What were the two choices? (e.g., control vs. wheat grass)

“*They were neither attracted nor repelled by a lower concentration of 1-hexanol, but were moderately attracted to a low concentration of hexanal. These results establish that hatchlings have a strong, innate preference for food odor blend, but the valence of the blend’s individual components may be different and may change depending on the concentration*”

What were the concentrations in “lower” and “low”? 

Need to mention the tracking device that was used to record the movements of locusts

**Introduction**

Expansion in the introduction section is required. Explanation of what kind of chemicals *Schistocerca americana* (or other locust species) are attracted or repelled, what method was used, and which life stage or phase was it. Moreover, the introduction does not discuss the effect of cooling and antenna amputation on odor perception in locusts, so this needs to be mentioned.An explanation of the “*new software toolkit for recording and tracking* (Line 48)” is required. What is the name? How does it work? What behaviors have been recorded in past studies?“*We found that even without any prior exposure to food, hatchlings could navigate to the source of a complex food odor blend, wheat grass juice. Notably, while attracted to the complex food blend, hatchlings were repelled by even low concentrations of the major component of the blend, 1-hexanol, presented alone. We also found the hatchlings were attracted by low concentrations of hexanal, which is both a food blend component and an aggregation pheromone but avoided it at higher concentrations. Thus, our results show that naïve hatchlings are innately attracted by food odor, that they can navigate to its source, and that individual monomolecular components of an attractive food odor can change valence depending on concentration*.”

These sentences should be in the discussion but not in the introduction. Rather than writing about the results from the study, questions asked in the study, prediction and reasoning behind the prediction should be included. 

**Method **

Lines 99-108 in the subsection *Behavior experiments* should be placed at the beginning of the method as it explains the information on insect husbandry. It also needs to mention where the insects were collected (or purchased) and the sample size for each experiment. Also, in what conditions (temperature, humidity, light cycle) was the wheat grown? Where were the wheat seeds purchased or collected from?Was the arena placed outside or inside that lab? Where (geolocation, name of the institute) was it conducted? Need to clearly state what behavioural signals that you have used to observe attraction/repulsion between control vs. food smells, with or without antenna, with or without coolingI think it would be better if you combine the subsections *Behavior arena*, *Odor delivery* and *Odor intensity map in arena* because they all explain the arena setups and refer to the same figureTo what concentrations the odorants were diluted? Need to mention in the method section too (not just in the result section)What chemicals were used to test olfactory capability in locusts with or without antennae or with or without cooling?Information on survival probability (Figure 2D) is missing. The method used to measure survival rate (Kaplan-Meier method) was first mentioned in the result section. Abbreviations in figure legends must be in the full name (e.g., PID in Figure 1, ROI in Figure 2). What do circles with different sizes indicate in Figure 1b & c?

**Result **

Please remove all “(see Materials and Methods)”What are circular paths?Sentences with references have to be moved into the introduction or discussion section (e.g., Line 190 “…earlier experiments’; Line 201-205 From “The major monomolecular volatile…” until “antennal lobes (Stopfer et al, 2003)”.The subheading “*Attraction to food odor is mediated by antennae*” is contradicting as the locusts are still attracted towards the food odor without antennae. It would be clearer if you could compare attraction to odor source between locusts with or without antennae (also for the cooling effect analysis; compare attraction between with or without cooling)

**Discussion**

Line 241 “…, *a plant readily consumed by locusts of all ages*” 

Reference?

Line 241 “*We also found that hatchlings require their antennae to navigate towards the odor source*”

Again, this is contradicting as hatchlings without antennae are attracted towards the smells. Also, it would be good to discuss past studies that have explored odor perception in grasshoppers after antenna amputation and cooling

Odor perception in mice and *Drosophila* are discussed but it would be better to focus on locusts which are well-studied in the past. For example, response to the wheatgrass smells and to green leaf volatile was observed in locust hatchlings in this study, but how about in adults of *S. americana*? Or in other locust species (like *Locusta migratoria* and *Schistocerca gregaria*)? The idea Line 255 “*The diet of mother mice has been shown to influence olfactory neurodevelopment and odor preference in the newborn (Todrank et al., 2011). This possibility could be tested in locusts by manipulating the mother’s diet or the volatiles present in egg pods*” is very interesting so expansion of this idea would be good for discussion “Low” or “lower” concentration is very vague and needs to be clear about what concentrations were used in this study and what concentrations (with which solvents) have been tested in past studies.

Reviewers' comments:

Reviewer's Responses to Questions

**Comments to the Author**

1. Is the manuscript technically sound, and do the data support the conclusions?

Reviewer #1: Yes

Reviewer #2: Partly

2. Has the statistical analysis been performed appropriately and rigorously? 

Reviewer #1: Yes

Reviewer #2: I Don't Know

3. Have the authors made all data underlying the findings in their manuscript fully available?

Reviewer #1: Yes

Reviewer #2: Yes

4. Is the manuscript presented in an intelligible fashion and written in standard English?

Reviewer #1: Yes

Reviewer #2: No

5. Review Comments to the Author

Reviewer #1: The authors test the affinity of hatched locusts for wheatgrass odor blend and for individual components of the blend, 1-hexanol and hexanal. As measured by number of entries into the region containing wheatgrass and staying time within the region, the authors find that hatched locusts are attracted to the wheatgrass odor compared to control, and that this effect vanished when the antennae are removed, indicating olfactory-based attraction and navigation. Interestingly, the authors that individual components of the odor blend can have a repellent effect.

The methodology is appropriate, with appropriate controls and statistical analysis.

Reviewer #2: The current study has a good sample size, and I was very intrigued and impressed with the variety of observations that the authors have made. However, many details are missing and unclear in the abstract, introduction, and methods and it is not obvious what the authors have studied until the result section. Moreover, information is scattered in the method and therefore better organization in the method section is required. It was hard to understand how the tracking device works (which is the main observation tool used in the study) so a detailed explanation in the introduction/method sections is required. Finally, the authors need to mention whether the locusts are in solitarious or gregarious phases as the odor capacity changes between the two phases.

6. PLOS authors have the option to publish the peer review history of their article (what does this mean?). If published, this will include your full peer review and any attached files.

Reviewer #1: No

Reviewer #2: No

---

## [Author Response · Author response to Decision Letter 0]

7 Jun 2023

We thank the editors and reviewer for thoughtful and helpful comments that have helped us dramatically improve our manuscript. We include a detailed rebuttal letter along with this submission.

---

## [Editor Report · Decision Letter 1]

27 Jun 2023

Innate attraction and aversion to odors in locusts

PONE-D-23-10041R1

Dear Dr. Stopfer,

We’re pleased to inform you that your manuscript has been judged scientifically suitable for publication and will be formally accepted for publication once it meets all outstanding technical requirements.

Kind regards,

Efthimios M. C. Skoulakis, PhD

Academic Editor

PLOS ONE
---

## [Editor Report · Acceptance letter]

30 Jun 2023

PONE-D-23-10041R1 

Innate attraction and aversion to odors in locusts 

Dear Dr. Stopfer:

I'm pleased to inform you that your manuscript has been deemed suitable for publication in PLOS ONE. Congratulations! Your manuscript is now with our production department. 

Kind regards, 

on behalf of

Dr. Efthimios M. C. Skoulakis 

Academic Editor

PLOS ONE